# The prevalence and risk factors associated with Iron, vitamin B12 and folate deficiencies in pregnant women: A cross-sectional study in Mbeya, Tanzania

Sauli E. John[1], Kaunara Azizi[1], Adam Hancy[1], Abela Twin'omujuni[1], Doris Katana[1], Julieth Shine[1], Vumilia Lyatuu[1], Abraham Sanga[2], Ramadhani S. Mwiru[2], Fatma Abdallah[1], Geofrey Mchau[1], Tedson Lukindo[1], Analice Kamala[1], Patrick Codjia[2], Germana H. Leyna[1,3], Ray M. Masumo[1]*

1 Tanzania Food and Nutrition Centre (TFNC), Dar es Salaam, Tanzania, 2 The United Nations Children's Fund (UNICEF) Tanzania, Dar es Salaam, Tanzania, 3 Department of Epidemiology and Biostatistics, Muhimbili University of Health and Allied Sciences (MUHAS), Dar es Salaam, Tanzania

* rmasumo@yahoo.com

**Data Availability Statement:** All datasets underlying this study are freely available at the public repository https://osf.io/7ysb9/.

## Abstract

Maternal nutrition is an important forecaster of infant's and mother's health status in most developing countries. This study aimed at assessing the prevalence and associated risk factors of iron, vitamin B12, and folate deficiencies among pregnant women in Mbeya Tanzania. A cross-sectional study using a cluster randomized sampling was conducted among 420 pregnant women. A structured questionnaire was used to collect socio-demographic and dietary assessment. Body iron store was assessed using serum ferritin measured by immunoturbidimetric assays using a Roche Cobas 400+ biochemistry analyzer. Serum folate was measured by folate microbiological assay, while serum vitamin B12 was measured by immunochemiluminescence assay using a Roche Cobas e411 immunoassay analyzer. Multivariate analysis was performed using Poisson regression. The prevalence of iron, folate, and vitamin B12 deficiencies among pregnant women in Mbeya was 37.8%, 24.0%, and 9.7% respectively. Higher odds of iron deficiency were seen in pregnant women aged 20–24 years older [Adjusted OR = 1.20 (95%CI 1.03, 1.35)], not employed [Adjusted OR = 3.0(95%CI 1.03–1.77)] and, not received iron/folic acid supplementation [Adjusted OR = 1.11 (95%CI 1.003–1.23)]. Pregnant women with highest and middle socio-economic statuses had lower odds of vitamin B12 deficiency [Adjusted OR = 0.83 (95%CI 0.76–0.92)] and [Adjusted OR = 0.89 (95%CI 0.81–0.98)] respectively. Pregnant women who were not employed, not received iron and folic acid supplement during pregnancy and, not consumed edible vegetable cooking oil had significant higher odds of serum folate deficiency [Adjusted OR = 3.0 (95%CI 1.58–5.68)], [Adjusted OR = 1.53 (95%CI 1.21–1.93)] and, [Adjusted OR = 2.77 (1.03–7.44)] respectively. This study confirms that iron, folate and vitamin B12 deficiencies are still a major challenge among pregnant women in Tanzania. We recommend for public health interventions for the provision of vitamin B12 along with iron and folic acid supplementations,

**Funding:** The authors received no specific funding for this work.

**Competing interests:** The authors have declared that no competing interests exist.

especially in pregnant women belong to low socio-economic status and limited knowledge of healthy diet.

## Introduction

Pregnancy is a unique period for a woman with a substantial impact on the health and well-being of both mother as well as the neonate [1]. Pregnancy is associated with increased demands of micronutrients due to the mother's own increased metabolic demands and also the development of the placenta and the fetus [2, 3]. Deficiency of micronutrients such as iron, folic acid, and B12, during pregnancy, is associated with anaemia and hypertension, besides impairing fetal function, development and growth [1–3].

Worldwide, iron deficiency is the most prevalent nutrient deficiency, estimated to affect 38.2% of pregnant women with the highest prevalence in South-East Asia (48.7%) and Africa (46.3%) [3, 4]. The prevalence of anaemia among pregnant women in Tanzania is 57%, making it a severe public health problem defined by the World Health Organization [5]. Deficiency of iron contributes more than two-thirds of all anaemia cases reported in Tanzanian population survey (TDHS 2010) [6]. However, iron deficiency is not the only nutrient that has been associated with anaemia in pregnancy, vitamin B12 and folate deficiencies also may contribute to increased anaemia prevalence [4, 7]. The previous TDHS missed the data on the prevalence of vitamin b12 and folate deficiencies.

According to the World Health Organization, the prevalence of vitamin B12 and folate deficiencies is perhaps a public health concern affecting millions of people globally [7]. However, there are few published studies and, the prevalence of folate and vitamin B12 deficiencies remains undefined [7]. Vitamin B12 is a micronutrient vital for cellular growth, differentiation, and development [8]. Further, Vitamin B12 and folate are essential for synthesizing DNA, RNA, lipids, and protein in the cellular cytoplasm [8, 9]. Specifically, vitamin B12 and folate are essential co-factors for the conversion of homocysteine to methionine required for the synthesis of neurotransmitters and phospholipids [8].

Animal proteins such as meat, eggs, and milk are the main dietary sources of vitamin B12 which plays a vital role in the folate metabolism and its deficiency predisposes to secondary folate deficiency which is specifically crucial in the rapidly dividing placental and fetal tissues [7]. In 2018, the Food and Agriculture Organization (FAO) estimated the meat supply in sub-Saharan African countries below 20 and this signifies the risk of vitamin B12 deficiencies [10]. Folate in pregnancy is essential in embryonic formation, particularly at the time of neural tube closure and in the prevention of birth defects and growth retardation [11]. Previous studies have linked insufficient folate during pregnancy with an increased risk of stillbirth, preterm delivery, and low birth weight [7].

A study conducted in Pakistan reported 6.8% of the adult population had vitamin B12 deficiency, and 39.7% had folate deficiency [12]. In Brazil, 0.3% of pregnant women had folate deficiency, and 7.9% had vitamin B12 deficiency [13]. A systematic review of the worldwide prevalence of vitamin B12 deficiency documented that, vitamin B12 deficiency is common during pregnancy even in non-vegetarian populations [14]. In sub-Saharan African countries and Tanzania in particular, there is a dearth of data on the epidemiology of iron, folate, and vitamin B12 deficiencies, as well as associated socio-demographic risk factors among pregnant women. The World Health Organization (WHO) recommends daily oral iron and folic acid supplementation with 30mg to 60mg of elemental iron and 400g (0.4mg) of

folic acid for pregnant women; however, vitamin B12 supplementation received little attention from public health perspectives [15]. Tanzania follows the WHO recommendations on supplementation of iron and folic acid during pregnancy for at least six months starting early at the first antenatal visit [15, 16]. Yet, adherence to iron and folic acid supplements during pregnancy is still low [16]. Evidence demonstrates that pregnant women in Tanzania taking iron and folic acid supplementation for at least 90 days were only twenty-one percent [16]. Nevertheless, the status of vitamin B12 in Tanzania is also low owing to poor consumption of animal protein and further, there is no mandatory recommendation for vitamin B12 supplementation during pregnancy [17]. Studies have demonstrated positive pregnancy outcomes like placental health and function, birth weight, and other related biomarkers with the supplementation of vitamin B12 along with iron and folic acid [18]. Therefore, the present study aimed to assess the prevalence and associated risk factors of iron, vitamin B12, and folate deficiencies among pregnant women in Mbeya Tanzania.

## Materials and methods

### Ethical consideration

Ethical clearance was obtained from the National Institute for Medical Research (NIMR), Ethical Review Committee with Reference Number SZECH-2439/R.A/V.1/49. Written consent was obtained from all study participants above 18 years of age. For the participants under 18 years of age, written informed consent was also obtained from their parents/guardians. All procedures followed were per the ethical standards of the Helsinki Declaration of 1975 [19].

### Study setting and design

This cross-sectional study was carried out among pregnant women gestation age below 28 weeks) attending antenatal clinics in Mbeya region Tanzania as part of epidemiological research project entitled 'the Improving Maternal and Adolescent Nutrition' (IMAN) in 2020. Mbeya is located in the southern highlands of Tanzania and lies at an altitude of 475metres above sea level with high peaks of 2981metres above sea level in Rungwe district.

### Study population

The study population comprised all pregnant women with less than 28 weeks of gestation and who attended antenatal clinics in the Reproductive and Child Health (RCH) clinics in the Mbeya region between September and October 2020. The population of Mbeya region was 2,707,410 as of the 2012 Tanzania national census [20]. This study was conducted in 42 RCH clinics in the seven districts which constitute Mbeya region. The region had 318 health facilities (both government, private and faith-based organizations) of which 251 provided RCH services.

### Sample size and sampling procedure

A pre-calculated sample size of 574 pregnant women with less than 28 weeks of gestation was estimated assuming a significance level of 5%, a 95% confidence intervals and a design factor of 1.5. An additional 10% was added to the sample to account for pregnant women who refused to consent and those who were unable to communicate due to illness or taking medication were excluded from the study non-responders based on the Lwanga and Lemeshow formula on determining the minimum sample size needed to obtain statistically valid [21]. This study managed to obtain a total sample of 420 pregnant women which was considered satisfactory with low sampling error and small increased of confidence intervals width (95% CI). This

study employed a two steps sampling procedure: First, a list of all health facilities providing RCH services in the Mbeya region was obtained (n = 251) and used in a random selection of the health facilities to be involved in the study from each district. Based on the sampling frame of health facilities, a probability proportional to size was performed to allocate the number of facilities per District for inclusion in the survey [22]. Out of 251 health facilities, forty-two health facilities were randomly selected for the study. And the addition of two reserved clusters was included in the survey. Therefore, a total of 44 health facilities offering RCH services were visited and surveyed [22].

The second step involved the selection of pregnant women from each selected health facility. An eligibility form was used to list all pregnant women attending RCH services in the selected health facility. The resulting list of pregnant mothers served as the sampling frame for the selection of participants who met the inclusion criteria. Systematic random sampling was carried out using the list of mothers to randomly select the required pregnant women for each facility to participate in the survey based on probability and proportion to size sampling for the specific facility [22].

## Data and sample collection

Trained research assistants elicited the socio-demographic and dietary information of study participants using a pre-tested semi-structured questionnaire and conducted anthropometric measurements. The project developed an operations manual for blood collection, handling, and transportation of blood and urine samples. In each health facility, a temporary laboratory was set to facilitate sample collection and spot testing of malaria and Hb. A trained nurse collected about 10ml of blood sample through vein puncture from each consented study participant. The first sample of 3ml of whole blood samples was collected in vacutainer with EDTA anticoagulant (Purple top), the second sample of 3ml of blood was collected in vacutainer with Lithium Heparin anticoagulant (Green top) and the third sample of 4ml of blood in a vacutainer tube without any anticoagulant (Red top). All samples were collected under aseptic conditions and protected from direct sunlight for laboratory assessment of blood micronutrient biomarkers. Blood in a vacutainer tube without anticoagulant (Red top) was allowed to clot for 15-30min. Both whole-bloods in EDTA and Heparin vacutainer tubes were mixed thoroughly soon after collected to prevent clotting. EDTA blood was used for testing of malaria and Hb. All tubes were maintained at 4–8˚C for less than 2 hours before shifting to the temporary intermediate laboratory for further processing. At the temporary laboratory the samples were processed and aliquoted into cryovials. Precisely, 100microliter (μl) of whole blood in each EDTA tube was collected and mixed with ascorbic acid to make cell lysate samples for RBC folate determination. A fresh ascorbic acid diluent was prepared by dissolving 0.5g of ascorbic acid in 50 ml of distilled water. RBC folate samples were then prepared by diluting 1 part of fresh EDTA whole blood (100 μl) with 10 parts of 1 g/dl (1%) ascorbic acid solution (1ml), corresponding to 1/11 dilution. The mixture was then allowed to stand in the dark for 30 min at room temperature, and the hemolysate was promptly frozen (at −20 ˚C) to maintain the folate in reduced states. The clot from the non-anticoagulated (red top) vacutainer tube was removed by centrifuging at 2,000xg for 10 minutes. The resulting serums were separated and aliquoted into vials and stored at −20 ˚C. Processed red cell hemolysate, plasma, and serum aliquots were placed into a ≤-20˚C freezer for temporary storage. Each month, samples were transported at <-20˚C to the Tanzania Food and Nutrition Centre (TFNC) laboratory in Dar es Salaam and were stored at −80˚C until analysis. Samples storage was done in a monitored -80˚C freezer until they were analyzed. A designated laboratory technician monitored the temperature each day and keep records using the "Freezer Temperature Monitoring Form".

**Blood tests for serum ferritin, vitamin b12 and folate.** Levels of serum folate were measured at the TFNC laboratory using CDC folate microbiological assay protocol [23, 24]. Vitamin B12 was also measured at the TFNC laboratory by using standard electrochemiluminescence immunoassay (ECLIA) based on Roche Cobas e411 immunoassay platform (Roche Diagnostics, Germany). Serum Ferritin levels were measured by standard immunoturbidimetric assays using a Roche Cobas 400+ biochemistry analyzer (Roche Diagnostics, Germany). To ensure quality of its laboratory data, TFNC laboratory is registered and participate in Vitamin A laboratory- external quality assurance (VITAL EQA) program, Performance verification program for folate microbiological assay and Performance verification program for serum micronutrients all offered by Centres for Disease control and prevention (CDC), Atlanta, USA. As recommended by the WHO, ferritin assessment was accompanied by the analysis of acute-phase proteins to detect the presence of infection or inflammation which tends to increase the levels of serum ferritin [25]. To adjust for increased serum ferritin due to sub-clinical infection and other inflammatory conditions, we utilized arithmetic correction factor (CF) as proposed by Thurnham and colleagues [26]. In this approach, CF of 0.67 was applied only for samples that had evidence of inflammation (CRP>5 mg/L), and a cut-off of <15μg/L was then applied to the adjusted ferritin levels.

**Dietary assessment.** A standard structured questionnaire for dietary assessment constructed in English was translated in several steps from English into Kiswahili (a local language) by bi-lingual Kiswahili/English professionals, and then back-translated to English by independent translators into Kiswahili. Kiswahili is the main language in Tanzania, spoken proficiently by almost 95% of the population. The research team reviewed the Kiswahili questionnaire for semantic and conceptual equivalence to the source version. Sensitivity to culture and selection of appropriate words were considered. The structured questionnaire was piloted to a separate group of women (not part of this study) to evaluate the quality of the translations in terms of comprehensibility, readability and relevance to assess face validity. First, the demographic and socio-economic variables considered were: age of pregnant woman, gestational age, parity, marital status, education level, occupation status, household assets, number of visits to RCH, smoking and alcohol consumption. Second, the dietary assessments: Pregnant women were asked 'from when you woke up yesterday till you woke up this morning did you consume the following food items: dark green leafy vegetables, fruits, nuts and seeds, vegetable liquid oils and, white roots and tubers?' Responses were in a likert scale: (1) not at all consumed; (2) consumed once; (3) consumed twice and; (4) consumed thrice and more.

**Anthropometric measurements.** Weight was measured to the nearest 0.1kg with a battery-powered electronic scale (Seca, Hamburg, Germany) and height was measured to the nearest 0.1cm with a height model recommended by UNICEF [27]. Height was measured when the subject was not wearing shoes or a head covering. Mid Upper Arm circumference (MUAC) was also assessed to pregnant women using MUAC tapes.

## Statistical analysis

The data was collected and entered into the personal computer. Statistical analysis was done using Statistical Package for Social Sciences (SPSS/version 21) software. For the continuous variables of serum folate, vitamin B12 and ferritin (iron), means ± standard deviation (SD) was calculated. For the categorical variables, the frequency and percentages were calculated and the prevalence of serum folate, vitamin B12 and ferritin (iron) deficiencies were estimated. According to the WHO recommendations for assessing folate status in population [25], in this study, folate deficiency was defined as serum folate levels of <10nmol/L (<4ng/mL). Vitamin

B12 deficiency was defined as serum vitamin B12 levels of <203pg/mL (150pmol/L) [28]. The corresponding low level of serum ferritin (iron was defined as <15μg/L [25].

The predictors of serum levels of folate, vitamin B12 and iron deficiencies were analyzed using the Chi square test and Poisson regression at the univariate level. All variables which were significant P<0.20 at the univariate level were included in multivariate Poisson regression with the 95% confidence interval (CI) for adjusting confounders and effect modifier. All P values were two-sided and, P<0.05 was considered statistically significant.

### Reliability of dietary assessment

The internal consistency reliability scales were examined using Cronbach's alpha. Test-retest reliability analysis was performed using kappa statistics and Intra class correlation coefficients (ICC). The average agreement between the research assistants and the gold standard (PIs) on dietary assessment was Cohen's kappa 0.62. Duplicate interviews were performed with 20 randomly selected pregnant women a day apart. Test-retest reliability of reports on dietary assessment in terms of ICC was 0.72 (95% CI 0.64–0.78). Thus, acceptable levels of intra-interviews agreement (kappa >0.60) were obtained [29].

### Results

The mean age of study participant was 25.49 ± 6.37years. The prevalence of iron deficiency was 37.8% (n = 159), folate deficiency was 24.0% (n = 101) and, vitamin B12 deficiency was 9.7% (n = 41) among 420 pregnant women attending antenatal clinics in the Mbeya region and respectively (Table 1). The means and standard deviation (SD) of ferritin level (μg/L) was 16.3±4.5, folate (nmol/L) was 9.7±4.2 and, vitamin B12 (pg/mL) was 147.2±32.5 respectively.

The distributions of socio-economic, demographic and dietary characteristics of study participants were summarized in Table 2. Study participants were pregnant women of age range from 15 to 49 years, with one-third aged between 20 and 24 years. Among pregnant women aged between 20 and 24, almost half (45.5%) had iron deficiency. About one-fifth (20%) were in households with highest (richest) socio-economic status and among them, fourteen (16.9%), twenty six (31.3%) and, sixteen (18.8%) had vitamin B12, iron and, folate deficiencies respectively. Almost two-thirds (71.7%) had attended primary education, among whom 37.8% had iron deficiency and 25.9% had folate deficiency respectively. More than a half (62.8%) of pregnant women in Chunya district council had iron deficiency. About half of pregnant women visited antenatal clinics (ANC) at the frequency of only two to three times.

More than a half (51%) of the study participants reported to take Fansidar (SP) during current pregnancy and 36.9% of pregnant women reported that they received iron/folic acid supplementations. Among pregnant women who received iron/folic acid supplementation, thirty one (11.9%) had vitamin B12 deficiency and, seventy two (27.2%) had folate deficiency. As presented in Table 2, about one-quarter (24.7%) of the participants consumed dark leafy green

**Table 1. The prevalence of serum ferritin, vitamin B12, and folate deficiencies among pregnant women in Mbeya Tanzania (n = 420).**

| Serum Ferritin (μg/L) | Normal | 61.5 (261) |
|---|---|---|
| | Deficiency | 37.8 (159) |
| Serum Vitamin B12 (pmol/L) | Normal | 88.8 (379) |
| | Deficiency | 9.7 (41) |
| Serum Folate (nmol/L) | Normal | 76.0 (319) |
| | Deficiency | 24.0 (101) |

**Table 2. Socio-economic, demographic, and dietary characteristics of study participants in Mbeya Tanzania (n = 420).**

| Variables | Category | Total % (n) | Vitamin B12 deficiency % (n) | Iron (Ferritin) deficiency % (n) | Folate deficiency % (n) |
|---|---|---|---|---|---|
| Age group (years) | 15–19 | 19.5 (82) | 12.3 (10) | 35.4 (29) | 14.6 (12) |
| | 20–24 | 31.7 (133) | 9.8 (13) | 45.5 (60) | 29.3 (39) |
| | 25–29 | 23.6 (99) | 10.2 (10) | 40.2 (39) | 25.3 (25) |
| | ≥30 | 25.2 (106) | 7.8 (8) | 29.2 (31) | 23.6 (25) |
| Marital Status | Married | 56.7 (238) | 10.3 (24) | 34.6 (82) | 26.1 (62) |
| | Cohabit | 31.7 (133) | 9.1 (12) | 42.4 (56) | 21.6 (29) |
| | Single/ Divorced | 11.7 (49) | 10.2 (5) | 42.9 (21) | 20.4 (10) |
| Occupation | Formal employment | 3.6 (15) | 0.0 (0.0) | 60.0 (9) | 53.8 (8) |
| | Self-employed | 84.5 (355) | 10.0 (35) | 37.0 (131) | 23.6 (84) |
| | Not employed | 11.9 (50) | 12.0 (6) | 38.8 (19) | 18.0 (9) |
| Household wealth index | Highest—richest | 20 (84) | 16.9 (14) | 31.3 (26) | 18.8 (16) |
| | Richer | 20 (84) | 10.8 (9) | 37.3 (31) | 32.1 (27) |
| | Middle | 20 (84) | 13.3 (11) | 38.1 (32) | 26.2 (22) |
| | Poorer | 20 (84) | 6.0 (5) | 50.0 (42) | 13.1 (11) |
| | Lowest—poorest | 20 (84) | 2.4 (2) | 33.3 (28) | 29.8 (25) |
| Education Status | No education | 8.1(34) | 14.7 (5) | 38.2 (13) | 2.9 (1) |
| | Primary | 71.7 (301) | 9.1 (27) | 37.8 (113) | 25.9 (78) |
| | Secondary and above | 20.2 (85) | 10.7 (9) | 38.8 (33) | 25.9 (22) |
| District councils | Chunya | 10.7 (45) | 11.1 (5) | 62.8 (28) | 17.8 (8) |
| | Mbeya | 23.0 (97) | 22.1 (21) | 20.6 (20) | 26.8 (26) |
| | Mbarali | 22.1 (93) | 5.5 (5) | 37.4 (34) | 11.8 (11) |
| | Kyela | 11.9 (50) | 0.0 (0.0) | 52.0 (26) | 38.0 (19) |
| | Rungwe | 16.4 (69) | 7.2 (5) | 48.5 (33) | 21.7 (15) |
| | Busekelo | 7.8 (33) | 9.4 (3) | 30.3 (10) | 27.3 (9) |
| | Mbeya city | 8.1 (34) | 6.1 (2) | 23.5 (8) | 38.2 (13) |
| Number of ANC visits during this pregnancy | First visit | 38.8 (163) | 3.7 (6) | 35.6 (58) | 27.4 (45) |
| | 2–3 visits | 53.8 (226) | 14.9 (33) | 37.5 (84) | 22.1 (50) |
| | More than 3 visits | 7.4 (31) | 6.5 (2) | 54.8 (17) | 19.4 (6) |
| MUAC | MUAC<23cm | 3.8 (16) | 6.3 (1) | 25.0 (4) | 18.8 (3) |
| | MUAC> = 23–< 33cms | 91.2 (383) | 10.3 (39) | 39.6 (151) | 24.0 (92) |
| | MUAC> = 33cms | 5 (21) | 4.8 (1) | 19.0 (4) | 28.6 (6) |
| Received Fansidar (SP) | No | 48.6 (204) | 3.5 (7) | 38.2 (76) | 26.9 (54) |
| | Yes | 51.4 (216) | 16.0 (34) | 38.1 (82) | 20.8 (45) |
| Received Iron/folic acid Supplementation | No | 63.1 (265) | 6.5 (10) | 46.5 (72) | 18.7 (29) |
| | Yes | 36.9 (155) | 11.9 (31) | 33.1 (87) | 27.2 (72) |
| Dark leafy green vegetables | Not at all | 34.0 (143) | 7.0 (10) | 35.9 (51) | 25.9 (37) |
| | One time | 37.1 (156) | 7.8 (12) | 44.5 (69) | 23.7 (37) |
| | Two times | 24.7 (104) | 16.8 (17) | 34.0 (35) | 24.0 (25) |
| | Three or more times | 4.3 (18) | 11.1 (2) | 22.2 (4) | 11.1 (2) |
| Fruits | Not at all | 72.9 (307) | 8.9(27) | 39.5 (121) | 22.6 (69) |
| | One time | 18.8 (79) | 11.7 (9) | 33.8 (26) | 30.4 (24) |
| | Two times | 6.9 (29) | 14.3(4) | 27.6 (8) | 24.1 (7) |
| | Three or more times | 1.4 (6) | 3.1(1) | 66.7 (4) | 33.3 (1) |
| Nuts and seeds | Not at all | 65.3 (275) | 8.9 (24) | 44.3 (117) | 20.7 (57) |
| | One time | 25.4 (107) | 12.4 (13) | 36.8 (39) | 29.0 (31) |

*(Continued)*

**Table 2.** (Continued)

| Variables | Category | Total % (n) | Vitamin B12 deficiency % (n) | Iron (Ferritin) deficiency % (n) | Folate deficiency % (n) |
|---|---|---|---|---|---|
| | Two times | 6.4 (27) | 11.1 (3) | 33.3 (9) | 33.3 (9) |
| | Three or more times | 2.8 (12) | 9.1 (1) | 46.5 (6) | 27.3 (3) |
| Poultry | Not at all | 92.4 (389) | 9.4 (36) | 41.9 (161) | 24.9 (97) |
| | One time | 5.0 (21) | 15.0 (3) | 35.0 (7) | 9.5 (2) |
| | Two times | 2.1 (9) | 11.1 (1) | 22.2 (2) | 22.2 (2) |
| | Three or more times | 0.5 (2) | 50.0 (1) | 50.0 (1) | 0.0 (0) |
| Vegetable edible liquid oils | Not at all | 11.2 (47) | 8.5 (4) | 48.9 (22) | 31.9 (15) |
| | One time | 31.8 (1340 | 11.2 (15) | 41.8 (56) | 26.9 (36) |
| | Two times | 44.4 (187) | 7.7 (14) | 42.1 (77) | 19.8 (37) |
| | Three or more times | 12.6 (53) | 13.0 (6) | 31.9 (15) | 25.5 (12) |
| Roots and tubers | Not at all | 44.7 (188) | 10.2 (19) | 44.4 (83) | 21.3 (40) |
| | One time | 38.5 (162) | 8.7 (14) | 39.0 (62) | 23.5 (38) |
| | Two times | 14.5 (61) | 13.8 (8) | 39.0 (23) | 37.7 (23) |
| | Three or more times | 2.4 (10) | 0.0 (0) | 30.0 (3) | 0.0 (0) |
| Meat | Not at all | 73.6 (310) | 12.4 (38) | 42.6 (130) | 23.5 (73) |
| | One time | 20.4 (86) | 3.6 (3) | 32.9 (28) | 23.3 (20) |
| | Two times | 5.9 (25) | 0.0 (0) | 52.0 (13) | 32.0 (8) |

vegetables two times daily, among whom 16.8%, 34.0% and 24.0% had serum vitamin B12, ferritin and, folate deficiencies respectively. The corresponding figures for fruit consumption, twenty nine (6.9%) pregnant women report consumed fruit two times daily, and among pregnant women who consumed fruits 14.3%, 27.6% and 24.1% had serum vitamin B12, ferritin and, folate deficiencies respectively.

To control for potential confounders in the analyses, the multiple regression analysis with all variables that were significant in univariate analysis (Table 2) was included in the model (Table 3), higher odds of serum ferritin deficiency were seen significant in pregnant women aged 20–24 years older [Adjusted OR = 1.20 (95%CI 1.03, 1.35)], not employed [Adjusted OR = 3.0(95%CI 1.03–1.77)], resident in Chunya district council [Adjusted OR = 1.24 (95%CI 0.99–1.55)], resident in Kyela district council [Adjusted OR = 1.26 (95%CI 1.02–1.56)], resident in Rungwe district council [Adjusted OR = 1.27 (95%CI 1.05–1.54)], had vitamin A deficiency [Adjusted OR = 1.13 (95%CI 1.03–1.24)] and, not received iron/folic acid supplementation [Adjusted OR = 1.11 (95%CI 1.003–1.23)].

Pregnant women from the high and middle socio-economic statuses (high and middle quintiles) had significant lower odds of having serum Vitamin B12 deficiency [Adjusted OR = 0.83 (95%CI 0.76–0.92)] and [Adjusted OR = 0.89 (95%CI 0.81–0.98)] respectively, while pregnant women who did not receive fansidar (SP) during pregnancy had significant higher odds of serum Vitamin B12 deficiency [Adjusted OR = 1.11 (95%CI 1.04–1.18)].

Pregnant women who were not employed and did not receive iron and folic acid supplement during pregnancy and did not at all consume edible vegetable cooking oil had significant higher odds of serum folate deficiency [Adjusted OR = 3.0 (95%CI 1.58–5.68)], [Adjusted OR = 1.53 (95%CI 1.21–1.93)] and, [Adjusted OR = 2.77 (1.03–7.44)] respectively. Corresponding findings, a significant lower odds of serum folate deficiency was detected among pregnant women who did not attend formal education and did not receive fansidar (SP) during pregnancy [Adjusted OR = 0.24 (95%CI 0.06–0.92)], [Adjusted OR = 0.78 (95%CI 0.78–0.99)] respectively. In addition, pregnant women who did not consume nuts and seeds at all or

**Table 3. Socio-demographic, economic, and dietary factors associated with vitamin B12, Iron (serum ferritin), and folate (serum) deficiencies among pregnant women in Mbeya Tanzania (n = 420).**

| Variable | Category | Vitamin B12 deficiency | | Iron (Serum Ferritin) deficiency | | Folate (Serum) deficiency | |
|---|---|---|---|---|---|---|---|
| | | Crude PR, 95% CI | Adjusted PR, 95%CI | Crude PR, 95% CI | Adjusted PR, 95%CI | Crude PR, 95% CI | Adjusted PR, 95%CI |
| Age | 15–19 years | 0.95 (0.87, 1.04) | 0.98 (0.90, 1.07) | 1.06 (0.92, 1.22) | 1.03 (0.90, 1.18) | 0.82 (0.58, 1.15) | 0.82 (0.59, 1.14) |
| | 20–24 years | 0.97 (0.90, 1.05) | 0.99 (0.91, 1.06) | 1.17 (1.04, 1.33)** | **1.20 (1.06, 1.35)**\*\* | 1.14 (0.85, 1.54) | 1.24 (0.93, 1.64) |
| | 25–29 years | 0.97 (0.89, 1.06) | 0.97 (0.89, 1.05) | 1.11 (0.97, 1.27) | 1.11 (0.98, 1.26) | 1.003 (0.72, 1.38) | 1.03 (0.76, 1.39) |
| | ≥30 years | 1 | 1 | 1 | 1 | 1 | 1 |
| Household wealth index | Highest-richest | 0.86 (0.79, 0.94)** | **0.83 (0.76, 0.92)**\*\* | 0.98 (0.84, 1.13) | 1.04 (0.89, 1.21) | 0.76 (0.53, 1.08) | 0.93 (0.65, 1.33) |
| | Richer | 0.91 (0.84, 1.01) | 0.93 (0.85, 1.03) | 1.04 (0.90, 1.20) | 1.05 (0.90, 1.22) | 1.19 (0.84, 1.69) | 1.29 (0.90, 1.86) |
| | Middle | 0.89 (0.82, 0.98)* | **0.89 (0.81, 0.98)**\*\* | 1.04 (0.90, 1.21) | 1.04 (0.90, 1.21) | 0.91 (0.64, 1.29) | 1.06 (0.75, 1.51) |
| | Poorer | 0.96 (0.88, 1.05) | 0.97 (0.89, 1.06) | 1.18 (1.02, 1.36)* | 1.14 (0.99, 1.31) | 0.65 (0.46, 0.92)* | 0.73 (0.51, 1.01) |
| | Lowest-poorest | 1 | 1 | 1 | 1 | 1 | 1 |
| Marital Status | Married | 0.99 (0.91, 1.09) | 1.02 (0.93, 1.12) | 0.92 (0.79, 1.06) | 0.96 (0.83, 1.11) | 1.18 (0.82, 1.69) | 1.29 (0.91, 1.82) |
| | Cohabit | 1.01 (0.91, 1.11) | 1.04 (0.94, 1.14) | 0.99 (0.85, 1.16) | 0.86 (0.86, 1.17) | 0.95 (0.65, 1.40) | 0.94 (0.65, 1.35) |
| | Single/Divorced | 1 | 1 | 1 | 1 | 1 | 1 |
| Occupation | Not employed | 1.12 (0.95, 1.33) | 1.16 (0.97, 1.37) | 1.23 (0.93, 1.63) | **1.35 (1.03, 1.77)**\*\* | 3.00 (1.54, 5.83)** | **3.00 (1.58, 5.68)**\*\* |
| | Self-employed | 1.02 (0.93, 1.11) | 1.04 (0.95, 1.14) | 0.98 (0.85, 1.13) | 0.94 (0.82, 1.09) | 1.09 (0.77, 1.53) | 1.09 (0.78, 1.53) |
| | Formal-employment | 1 | 1 | 1 | 1 | 1 | 1 |
| Education Status | No education | 0.86 (0.60, 1.22) | 0.90 (0.63, 1.29) | 1.05 (0.59, 1.86) | 0.83 (0.47, 1.45) | 0.13 (0.03, 0.54)** | **0.24 (0.06, 0.92)**\* |
| | Primary | 0.91 (0.65, 1.28) | 0.92 (0.65, 1.30) | 1.04 (0.60, 1.81) | 0.83 (0.48, 1.43) | 0.26 (0.07, 0.97)* | 0.39 (0.10, 1.43) |
| | Secondary | 0.89 (0.63, 1.26) | 0.87 (0.62, 1.23) | 1.05 (0.60, 1.85) | 0.81 (0.47, 1.38) | 0.24 (0.06, 0.93)* | 0.34 (0.09, 1.24) |
| | College and university | 1 | 1 | 1 | 1 | 1 | 1 |
| District Councils | Chunya | 0.95 (0.83, 1.08) | 0.97 (0.84, 1.12) | 1.47 (1.19, 1.81)** | **1.24 (0.99, 1.55)**\* | 0.56 (0.34, 0.94)* | 1.07 (0.63, 1.83) |
| | Mbeya | 0.85 (0.76, 0.95)** | 0.89 (0.78, 1.01) | 0.97 (0.81, 1.16) | 0.98 (0.81, 1.20) | 0.78 (0.49, 1.22) | 1.21 (0.76, 1.93) |
| | Mbarali | 1.01 (0.89, 1.12) | 1.01 (0.89, 1.14) | 1.14 (0.95, 1.37) | 1.08 (0.88, 1.31) | **0.46 (0.29, 0.72)**\*\* | 0.65 (0.41, 1.05) |
| | Kyela | 1.06 (0.93, 1.20) | 1.10 (0.96, 1.26) | 1.32 (1.08, 1.62)** | **1.26 (1.02, 1.56)**\* | 1.07 (0.65, 1.76) | 1.37 (0.82, 2.27) |
| | Rungwe | 0.98 (0.87, 1.11) | 1.04 (0.92, 1.18) | 1.28 (1.06, 1.55)** | **1.27 (1.05, 1.54)**\* | 0.70 (0.44, 1.13) | 0.90 (0.57, 1.43) |
| | Busekelo | 0.96 (0.84, 1.11) | 1.02 (0.88, 1.19) | 1.07 (0.85, 1.33) | 1.14 (0.90, 1.43) | 0.78 (0.45, 1.35) | 0.99 (0.57, 1.71) |
| | Mbeya city | 1 | 1 | 1 | 1 | 1 | 1 |
| Received Fansidar (SP) | No | 1.13 (1.07, 1.20)** | **1.11 (1.04, 1.18)**\*\* | 0.99 (0.90, 1.09) | 0.93 (0.84, 1.03) | 1.20 (0.96, 1.51) | **0.78 (0.78, 0.99)**\* |
| | Yes | 1 | 1 | 1 | 1 | 1 | **1** |
| Anaemia status | No | 0.99 (0.93, 1.06) | 1.03 (0.96, 1.10) | 0.74 (0.67, 0.820)** | 0.81 (0.73, 0.89) | 1.05 (0.83, 1.39) | 1.04 (0.82, 1.33) |
| | Yes | 1 | 1 | 1 | 1 | 1 | 1 |
| Vitamin A deficiency | Deficiency | 1.03 (0.97, 1.09) | 1.01 (0.95, 1.06) | 1.21 (1.10, 1.33)** | **1.13 (1.03, 1.24)**\*\* | 1.07 (0.86, 1.34) | 1.10 (0.89, 1.36) |
| | Normal | 1 | 1 | 1 | 1 | 1 | 1 |
| Received Iron/folic acid Supplementation | No | 1.05 (0.99, 1.12) | 1.01 (0.94, 1.07) | 1.14 (1.03, 1.25)** | **1.11(1.003, 1.23)**\* | 0.81 (0.64, 1.02) | **1.53 (1.21, 1.93)**\*\* |
| | Yes | 1 | 1 | 1 | 1 | 1 | 1 |
| Consumption of Vegetables | Not at all | 0.89(0.73, 1.09) | 0.92 (0.75, 1.13) | 0.83 (0.60, 1.14) | 0.84 (0.61, 1.15) | 1.16 (0.53, 2.51) | 0.78 (0.37, 1.68) |
| | One time | 0.89 (0.73, 1.10) | 0.92 (0.74, 1.13) | 0.82 (0.59, 1.15) | 0.84 (0.60, 1.16) | 1.46 (0.66, 3.24) | 0.86 (0.39, 1.87) |

*(Continued)*

**Table 3.** (Continued)

| Variable | Category | Vitamin B12 deficiency | | Iron (Serum Ferritin) deficiency | | Folate (Serum) deficiency | |
|---|---|---|---|---|---|---|---|
| | | Crude PR, 95% CI | Adjusted PR, 95%CI | Crude PR, 95% CI | Adjusted PR, 95%CI | Crude PR, 95% CI | Adjusted PR, 95%CI |
| | Two times | 0.94 (0.76, 1.16) | 0.95 (0.77, 1.18) | 0.86 (0.61, 1.22) | 0.90 (0.64, 1.26) | 0.91 (0.40, 2.08) | 0.56 (0.25, 1.25) |
| | Three or more times | 1 | 1 | 1 | 1 | 1 | 1 |
| Consumption of fruits | Not at all | 1.27 (0.91, 1.79) | 1.20 (0.86, 1.68) | 0.76 (0.44, 1.32) | 0.86 (0.51, 1.45) | 1.73 (0.46, 6.47) | 1.33 (0.38, 4.67) |
| | One time | 1.24 (0.88, 1.74) | 1.13 (0.80, 1.58) | 0.72 (0.41, 1.25) | 0.82 (0.48, 1.40) | 2.36 (0.62, 9.00) | 1.62 (0.46, 5.75) |
| | Two times | 1.21 (0.84, 1.72) | 1.13 (0.79, 1.60) | 0.67 (0.38, 1.20) | 0.78 (0.45, 1.35) | 1.90 (0.47, 7.56) | 1.44 (0.38, 5.37) |
| | Three or more times | 1 | 1 | 1 | 1 | 1 | 1 |
| Consumption of nuts and seeds | Not at all | 0.91 (0.51, 1.64) | 1.20 (0.86, 1.68) | 0.54 (0.21, 1.40) | 1.09 (0.36, 3.30) | 0.08 (0.01, 0.81)* | **0.01 (0.001, 0.19)**\*\* |
| | One time | 0.88 (0.49, 1.58) | 1.13 (0.80, 1.58) | 0.52 (0.20, 1.36) | 1.05 (0.35, 3.17) | 0.10 (0.01, 1.01)* | **0.01 (0.001, 0.22)**\*\* |
| | Two times | 0.89 (0.49, 1.62) | 1.13 (0.79, 1.60) | 0.49 (0.18, 1.29) | 1.08 (0.35, 3.28) | 0.12 (0.01, 1.21) | **0.01 (0.001, 0.23)**\*\* |
| | Three or more times | 1 | 1 | 1 | 1 | 1 | 1 |
| Consumption of white roots | Not at all | 0.90 (0.74, 1.10) | 0.89 (0.74, 1.08) | 1.24 (0.91, 1.69) | 1.14 (0.86, 1.52) | 1.70 (0.81, 3.54) | 1.43 (0.72, 2.82) |
| | One time | 0.91 (0.75, 1.11) | 0.90 (0.75, 1.10) | 1.17 (0.86, 1.59) | 1.05 (0.79, 1.41) | 1.83 (0.87, 3.82) | 1.38 (0.69, 2.74) |
| | Two times | 0.87 (0.70, 1.07) | 0.88 (0.72, 1.08) | 1.16 (0.84, 1.60) | 1.07 (0.79, 1.45) | 2.71 (1.25, 5.88)* | 1.93 (0.93, 3.98) |
| | Three or more times | 1 | 1 | 1 | 1 | 1 | 1 |
| Consumption of vegetable edible liquid oil | Not at all | 1.28 (0.99, 1.64)* | 1.20 (0.92, 1.56) | 1.33 (0.88, 2.01) | 1.27 (0.84, 1.92) | 1.48 (0.55, 3.96) | **2.77 (1.03, 7.44)**\*\* |
| | One time | 1.24 (0.97, 1.59) | 1.19 (0.93, 1.53) | 1.22 (0.82, 1.82) | 1.27 (0.86, 1.89) | 1.26 (0.48, 3.25) | 1.98 (0.77, 5.06) |
| | Two times | 1.29 (1.01, 1.64)* | 1.26 (0.98, 1.61) | 1.25 (0.84, 1.86) | 1.28 (0.87, 1.89) | 0.97 (0.38, 2.50) | 1.68 (0.65, 4.20) |
| | Three or more times | 1 | 1 | 1 | 1 | 1 | 1 |

Poisson regression: Reference group is the last category

*P<0.05

**P<0.01

did that in few times were significantly associated with lower odds of serum folate deficiency among pregnant women in Mbeya [Adjusted OR = 0.01 (95%CI 0.001–0.19)] and [Adjusted OR = 0.01 (95%CI 0.001–0.24)] respectively (Table 3).

## Discussion

Maternal nutrition plays an important role in both maternal and fetal health outcomes. Under-nutrition, micronutrient deficiencies, overweight and obesity during pregnancy are gaining increasing importance as major causes of ill-health worldwide [30]. About two billion people are affected by micronutrient deficiencies in the world and pregnant women are more affected [30]. However in Tanzania and most sub-Saharan African countries, there is a dearth of information concerning micronutrient deficiencies among pregnant women [31, 32]. The present study focused on assessing the prevalence and associated risk factors of iron, vitamin B12, and folate deficiencies among pregnant women attending antenatal clinics in Mbeya Tanzania.

Our study has shown that more than one micronutrients deficiencies among pregnant women are common in this community as was previously documented by Bailey and colleagues in 2015 [30] and, also by Yeneabat and colleagues in 2019 [33]. The fact that Tanzania has a broader coverage of antenatal services plays a robust platform for distributing tablets of iron and folic acid and implementing the World Health Organization's recommendations that all pregnant women to receive 90 or more tablets during pregnancy, however, the present study revealed that only 36.9% of pregnant women of gestation period of less than 28 weeks had received iron and folic acid supplementation.

In 2012, Olukemi Ogundipe and colleagues documented that out of 21,889 women who delivered at Kilimanjaro Christian Medical Centre (KCMC) between 1999 and 2008, only 16% of pregnant women reported intake of iron and folic acid supplementation [34]. Slightly similar findings were documented by Lyoba and colleagues in 2020, out of 320 study participants only 20.3% adhered to iron and folic acid supplementation [16]. The proportion of pregnant women using folate and iron supplementation in the present study was slight similar to the reported usage in Kenya (32.7%) [35]. These differences could be attributed by different study subject, socio-demographic characteristics and access of quality reproductive and child health services offered in the health facilities.

A systemic review of vitamin B12 deficiency among pregnant women in all trimesters documented prevalence in a range of 20%–30% [14]. Our study found a slightly higher prevalence of vitamin B12 deficiency as compared to the study of Brazilian pregnant women who attended public health centers [13]. The difference between these two studies could be attributed to the differences in the assays employed. Further, our analysis revealed that, the highest socioeconomic status was attributed to the low odds of vitamin B12 deficiency among pregnant women as was early documented by Shobha and colleagues [36]. Most of our subjects reported low consumption of meat and poultry and a similar finding was documented by Pawlak and colleagues in their comprehensive review of the literature which showed a relatively significant deficiency prevalence of vitamin B12 among the vegetarians [37]. Interestingly, in the present study, we reported that vitamin B12 deficiency was significantly associated with pregnant women who did not receive fansidar (Sulfadoxine/pyrimethamine, SP) tablets and this calls for further investigations to understand this association.

We also found that about a quarter of the pregnant women in Mbeya region of Tanzania had serum folate levels below 3nmol/L, which revealed that the prevalence of folate deficiency is among the leading nutritional problems of public health concern in Tanzania. The present study, as was demonstrated in other studies, has managed to show pregnant women who do not take iron and folic acid supplementation that are not employed and hence have lower socioeconomic status, and those who are on restricted diets especially consumption of edible vegetable oil are at the risk of folate deficiency [38–42]. Studies have shown that during pregnancy, the folate needs to be increased to maintain the requirement of the mother, the fetus, and placenta in the synthesis of protein, DNA, and RNA [43, 44]. Moreover, shreds of evidence have demonstrated that folate deficiency during pregnancy is detrimental to the infant and can predispose to chronic diseases later in life such as neural tube disorders, premature birth and, low birth weight [43, 44].

The World Health Organization (WHO) recommends serum ferritin concentrations as the best indicator of iron deficiency and, the serum ferritin concentrations below 15μg/L among pregnant women indicate iron deficiency [45, 46]. Studies on serum ferritin have reported an increased level of ferritin if inflammation is present [47]. In this study, we applied the CF [26] to get a more precise estimate of serum ferritin in all pregnant women who suspected to have inflammation. The present study found more than one third of pregnant women had iron deficiency as indicated by low serum ferritin level. Despite the low prevalence of folate deficiency

and vitamin B12 deficiency in this population but still the prevalence of iron deficiency was comparable high. In 2008, Metz documented that biochemical deficiency of folic acid and vitamin B12 does not translate into comparable prevalence of anaemia due to iron deficiency [48]. We also observe a higher risk of iron deficiency among pregnant women aged between 20–24 years old, not employed, resident in Chunya, Rungwe and Kyela district councils (as an indicative of lower socioeconomic status). Similar observation was documented by Haile and colleagues that hormonal contraceptive use (as an indicative of higher socioeconomic status) was significantly associated with a reduced risk of iron deficiency [49]. Furthermore, the present study found increased risk of iron deficiency among pregnant women who had vitamin A deficiency and also have not received iron or folic acid supplements. Absorption of dietary iron from the human gut depends on physiological requirements, but may be restricted by the quantity or bioavailability; Msemo and colleagues documented that high prevalence of iron deficiency without concurrent folate deficiency might be due to poor bioavailability rather than dietary insufficiency [35].

We believe that our results represent the profile of pregnant women with gestation period of below 28 weeks and, as the measurements were performed in 420 using laboratory methods considered suitable for analysis. In Tanzania and many developing countries, much effort has been given to iron and folic acid supplementation during pregnancy, but less emphasis has been given to vitamin B12 supplements. Vitamin B12 deficiency may result in secondary folate deficiency and, folate in pregnancy plays a vital role in embryonic formation [8]. Our findings underline the urgency to establish public health interventions that address micronutrient deficiencies during pregnancy. At present, during pregnancy in Tanzania, there is no guidance for the supplementation of vitamin B12. The potential benefit of multiple micronutrients supplementation has been demonstrated in the double-blind cluster-randomized trial on the effect of maternal multiple micronutrient supplementations on fetal loss and infant death in Indonesia, opens the intriguing probability that such intrauterine supplements may favour neonates beyond the neonatal period [18].

## Conclusions

One-third of pregnant women with a gestation period of below 28 weeks were deficient in iron and, about half were folate deficient and, about one-tenth were vitamin B12 deficient affirms that micronutrients such as iron, folic acid, and vitamin B12 are major public health challenges in Tanzania. In addition, our results enable the identification of key risk factors associated with folate, iron, and vitamin B12 deficiencies. We suggest that iron, folic acid, and micronutrients (including vitamin B12) supplementation programs should be strengthened, especially among pregnant women belonging to low socio-economic status. Furthermore, interventions to improve community uptake of micronutrient should be promoted especially to pregnant women belonging to low socio economic status, unemployed and limited knowledge of healthy diet. It is recommended that future longitudinal studies are needed to explore the risk factors of serum folate, ferritin and vitamin B12 deficiencies among pregnant women.

## Acknowledgments

First, we are grateful to pregnant women, health workers and all of those with whom we had the pleasure to work during this project. Second, we are appreciating the technical support provided by the Centre of Disease Control and Prevention (CDC), Atlanta, USA on all folate assays. Lastly we are appreciating the support and collaboration of the Muhimbili Orthopedic Institute (MOI), Tanzania for allowing its laboratory facility and staff to be used for the remaining laboratory assays in this study.

## Author Contributions

**Conceptualization:** Sauli E. John, Kaunara Azizi, Vumilia Lyatuu, Abraham Sanga, Ramadhani S. Mwiru, Fatma Abdallah, Geofrey Mchau, Tedson Lukindo, Patrick Codjia, Ray M. Masumo.

**Data curation:** Sauli E. John, Kaunara Azizi, Abela Twin'omujuni, Tedson Lukindo.

**Formal analysis:** Sauli E. John, Adam Hancy, Doris Katana, Julieth Shine, Analice Kamala, Ray M. Masumo.

**Funding acquisition:** Germana H. Leyna.

**Investigation:** Kaunara Azizi, Analice Kamala.

**Methodology:** Sauli E. John, Kaunara Azizi, Abraham Sanga, Tedson Lukindo, Ray M. Masumo.

**Project administration:** Sauli E. John, Abela Twin'omujuni, Vumilia Lyatuu, Abraham Sanga, Ramadhani S. Mwiru, Fatma Abdallah, Geofrey Mchau, Patrick Codjia, Germana H. Leyna.

**Resources:** Abraham Sanga, Germana H. Leyna.

**Software:** Adam Hancy, Julieth Shine.

**Supervision:** Abraham Sanga, Ramadhani S. Mwiru, Patrick Codjia, Germana H. Leyna, Ray M. Masumo.

**Validation:** Kaunara Azizi, Vumilia Lyatuu, Tedson Lukindo, Analice Kamala, Germana H. Leyna.

**Visualization:** Abela Twin'omujuni.

**Writing – original draft:** Sauli E. John, Kaunara Azizi, Adam Hancy, Doris Katana, Julieth Shine, Vumilia Lyatuu, Abraham Sanga, Fatma Abdallah, Geofrey Mchau, Tedson Lukindo, Ray M. Masumo.

**Writing – review & editing:** Sauli E. John, Ramadhani S. Mwiru, Analice Kamala, Patrick Codjia, Germana H. Leyna, Ray M. Masumo.

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
