## [Decision Letter · Decision Letter 0]

8 Feb 2023

PGPH-D-22-01831

The prevalence and risk factors associated with the deficiencies of Iron, Vitamin B12 and Folate among pregnant women attending antenatal clinics in Mbeya Tanzania

Dear Dr. Masumo,

Thank you for submitting your manuscript to PLOS Global Public Health. After careful consideration, we feel that it has merit but does not fully meet PLOS Global Public Health’s publication criteria as it currently stands. Therefore, we invite you to submit a revised version of the manuscript that addresses the points raised during the review process.

We look forward to receiving your revised manuscript.

Kind regards,

Dickson Abanimi Amugsi, PhD

Academic Editor

Journal Requirements:

1. In the ethics statement in the Methods, you have specified that verbal consent was obtained. Please provide additional details regarding how this consent was documented and witnessed, and state whether this was approved by the IRB.

a. Please clarify all sources of funding (financial or material support) for your study. List the grants (with grant number) or organizations (with url) that supported your study, including funding received from your institution. 

b. State the initials, alongside each funding source, of each author to receive each grant.

c. State what role the funders took in the study. If the funders had no role in your study, please state: “The funders had no role in study design, data collection and analysis, decision to publish, or preparation of the manuscript.”

d. If any authors received a salary from any of your funders, please state which authors and which funders.

3. In the Funding Information you indicated that no funding was received. Please revise the Funding Information field to reflect funding received.

4. Your manuscript is missing the following sections: Introduction. Please ensure these are present, and in the correct order, and that any references to subheadings in your main text are correct. An outline of the required sections can be consulted in our submission guidelines here:

https://journals.plos.org/globalpublichealth/s/submission-guidelines#loc-parts-of-a-submission

Additional Editor Comments (if provided):

Thank you for submitting your work to PGPH for publication. Two independent reviewers have reviewed your manuscript and recommended minor revision. However, I have noticed that the issues they raised are quite substantial, and will need some amount of effort to address. I suggest you diligently address their comments and resubmit your manuscript for consideration. Also, carefully proofread the entire manuscript to address possible grammar and diction issues.

I hope to receive your revise manuscript soon.

Thank you.

Reviewers' comments:

Reviewer's Responses to Questions

**Comments to the Author**

1. Does this manuscript meet PLOS Global Public Health’s publication criteria? Is the manuscript technically sound, and do the data support the conclusions? The manuscript must describe methodologically and ethically rigorous research with conclusions that are appropriately drawn based on the data presented.

Reviewer #1: Yes

Reviewer #2: Yes

2. Has the statistical analysis been performed appropriately and rigorously?

Reviewer #1: Yes

Reviewer #2: Yes

3. Have the authors made all data underlying the findings in their manuscript fully available (please refer to the Data Availability Statement at the start of the manuscript PDF file)?

Reviewer #1: No

Reviewer #2: Yes

4. Is the manuscript presented in an intelligible fashion and written in standard English?

Reviewer #1: Yes

Reviewer #2: No

5. Review Comments to the Author

Reviewer #1: TITLE

1. Clarify if the study focuses on one or all three of the deficiencies, Iron, Vitamin B12 and Folate.

2. Consider including a statement in the title about the geographical focus of the study (e.g. Mbeya, Tanzania).

3. Consider adding a phrase such as "a cross-sectional study" to provide additional information about the study design.

ABSTRACT

1. I recommend that the authors provide more detail regarding the methodology used in the study, such as the sampling technique and the specific dietary patterns that were assessed.

2. The authors should provide more clarity on the interpretation of the results. For example, it would be helpful to provide more information on how the different factors were associated with the deficiencies, and what implications these findings have for pregnant women in Tanzania.

3. The authors should consider extending their discussion to include implications for policy and public health interventions.

INTRODUCTION

I recommend that the authors should provide

1. More information about the prevalence of iron, vitamin B12, and folate deficiencies in other countries and regions, as well as a comparison to the prevalence in Tanzania.

2. A more detailed description of the potential health risks associated with deficiencies in iron, vitamin B12, and folate, including both maternal and neonatal risks.

3. More information about the current recommendations for vitamin B12 supplementation during pregnancy in Tanzania and the potential benefits of supplementing with vitamin B12.

METHODS

Sampling and data collection procedure

The authors should

1. Provide more information on the Lwanga and Lemeshow formula and its relevance to the study.

2. Provide further explanation on the two-stage cluster sampling used in the study.

3. Discuss the implications of the reduced sample size on the results of the study.

4. Ensure that the operations manual for blood collection, handling, and transportation of blood and urine samples is detailed and comprehensive.

5. Also provide more information on the exact procedures used for cell lysate sample preparation and storage.

6. Provide further information on the temperature and duration of the transport of the samples to the Tanzania Food and Nutrition Centre (TFNC) laboratory.

7. Should include more information on how the samples were handled and processed at the TFNC laboratory.

Study tool

1. It would be beneficial for the authors to provide more information on the translation process. This could include information on the qualifications of the translators and the methods used to check the accuracy of the translation.

2. The authors should consider making the questionnaire available in the paper for readers to review.

3. The authors should state the response categories for each question more clearly.

4. The authors should consider adding information on how they ensured the reliability and validity of the questionnaire within the context of their study and environment.

RESULTS

The authors have done a thorough job of presenting the data collected from their study. However,

1. It would be beneficial to provide more detailed descriptions of the socio-economic, demographic, and dietary characteristics of the study participants.

2. Provide information on the potential confounding factors that may have influenced their findings.

DISCUSSION

The authors should consider

1. Expanding on the implications of their findings and discussing potential strategies to address the identified deficiencies among pregnant women.

2. Exploring the potential benefits of multiple micronutrients supplementation, as well as the potential long-term implications of deficiencies during pregnancy.

3. Contextualizing their findings and exploring the potential differences in prevalence and associated risk factors across different regions of the country.

Reviewer #2: Review comments: The study done in Tanzania, it is an excellent study that brings out the factors that are associated with risk factors related to multiple micronutrient deficiencies and it is worthy to be published. However the manuscript needs minor reviews as it contains some repeats, grammar, typo’s and unnecessary long sentences, which may create loss of interest to the reader in getting the key massages from the study. The following are some of the areas that the author may consider to review:-

1. According to the guidelines of PLOS, the author is supposed to provide numbering for each line sentence of the manuscript to easy referencing by the reviewer. This was not provided. Consider incorporating the line numbering

2. Rephrase the sentence for clarity in the methodology: …Mbeya is located in the southern highlands of Tanzania and lies at an altitude of 475metres above sea level with high peaks of 2981metres above sea level at Rungwe district higher attitudes.

3. Refer to the sentence on the methodology “ Precisely, 100ml of whole blood in each EDTA tube was collected and mixed with ascorbic acid to make cell lysate samples for RBC folate determination”

4. Address the references sequencing. See reference [12] then [14], then [13], [43] then [15]

5. Insert reference: Ethical clearance para: All procedures followed were per the ethical standards of the Helsinki Declaration of 1975[ ].

6. In the results part, rephrase the sentence: As presented in table 2, about one-quarter (24.7%) of the participants consumed two times dark leafy green vegetables daily among whom 16.8%, 34.0% and 24.0% had serum vitamin B12, ferritin and, folate deficiencies respectively.

7. A systemic review of vitamin B12 deficiency among pregnant women in all trimesters documented prevalence in a range of 20% - 30% [39]. In contrary, our study found a low prevalence of vitamin B12 deficiency among pregnant women of gestation period less than 28 weeks

8. Studies have showed that the optimal folate required is essential during pregnancy to encounter the growing demands of both the mother and the fetus [26, 32].

9. In conclusion para it is not clear rephrase it… We believe that our results represent the profile of pregnant women with gestation period of below 28 weeks and, as the measurements were performed in 420 subjects using laboratory methods considered suitable for analysis; …(it clearly brings out what new findings…, that affirms what known or unknown massage to the reader?). Furthermore, our results enable the identification of key risk factors of developing folate, iron and vitamin B12 deficiencies.

10. In Tanzania and many developing counties, much effort has been given to iron and folic acid supplementation during pregnancy, but less emphasis has been given to vitamin B12 supplements;

11. Rephrase this sentence in the conclusion: It is recommended that future studies longitudinal should explore the risk factors of serum folate, ferritin and vitamin B12 deficiencies among pregnant women

6. PLOS authors have the option to publish the peer review history of their article (what does this mean?). If published, this will include your full peer review and any attached files.

**Do you want your identity to be public for this peer review?** For information about this choice, including consent withdrawal, please see our Privacy Policy.

Reviewer #1: No

Reviewer #2: **Yes: **Vincent D. Assey

---

## [Decision Letter · Decision Letter 1]

23 Mar 2023

The prevalence and risk factors associated with Iron, Vitamin B12 and Folate deficiencies in pregnant women: A cross-sectional study in Mbeya, Tanzania

PGPH-D-22-01831R1

Dear Dr Masumo,

We are pleased to inform you that your manuscript 'The prevalence and risk factors associated with Iron, Vitamin B12 and Folate deficiencies in pregnant women: A cross-sectional study in Mbeya, Tanzania' has been provisionally accepted for publication in PLOS Global Public Health.

Best regards,

Dickson Abanimi Amugsi, PhD

Academic Editor

Reviewer Comments (if any, and for reference):

Reviewer's Responses to Questions

**Comments to the Author**

1. If the authors have adequately addressed your comments raised in a previous round of review and you feel that this manuscript is now acceptable for publication, you may indicate that here to bypass the “Comments to the Author” section, enter your conflict of interest statement in the “Confidential to Editor” section, and submit your "Accept" recommendation.

Reviewer #1: All comments have been addressed

Reviewer #2: All comments have been addressed

2. Does this manuscript meet PLOS Global Public Health’s publication criteria? Is the manuscript technically sound, and do the data support the conclusions? The manuscript must describe methodologically and ethically rigorous research with conclusions that are appropriately drawn based on the data presented.

Reviewer #1: Yes

Reviewer #2: Yes

3. Has the statistical analysis been performed appropriately and rigorously?

Reviewer #1: Yes

Reviewer #2: Yes

4. Have the authors made all data underlying the findings in their manuscript fully available (please refer to the Data Availability Statement at the start of the manuscript PDF file)?

Reviewer #1: Yes

Reviewer #2: Yes

5. Is the manuscript presented in an intelligible fashion and written in standard English?

Reviewer #1: Yes

Reviewer #2: Yes

6. Review Comments to the Author

Reviewer #1: (No Response)

Reviewer #2: Satisfied with author’s response in addressing my previous concerns. I recommend for acceptance of this manuscript for publication

7. PLOS authors have the option to publish the peer review history of their article (what does this mean?). If published, this will include your full peer review and any attached files.

**Do you want your identity to be public for this peer review?** For information about this choice, including consent withdrawal, please see our Privacy Policy.

Reviewer #1: **Yes: **Dr. Calvince Anino

Reviewer #2: No
